# DNA Barcoding Analysis and Phylogenetic Relation of Mangroves in Guangdong Province, China

**Feng Wu** [1,2] , **Mei Li** [1], **Baowen Liao** [1,*], **Xin Shi** [1] and **Yong Xu** [3,4]

1    Key Laboratory of State Forestry Administration on Tropical Forestry Research, Research Institute of Tropical Forestry, Chinese Academy of Forestry, Guangzhou 510520, China; wufengcaf@163.com (F.W.); mangrovelimei@163.com (M.L.); hb-shixin@126.com (X.S.)

2    Zhaoqing Xinghu National Wetland Park Management Center, Zhaoqing 526060, China

3    Key Laboratory of Plant Resources Conservation and Sustainable Utilization, South China Botanical Garden, Chinese Academy of Sciences, Guangzhou 510650, China; xuyong@scbg.ac.cn

4    University of Chinese Academy of Sciences, Beijing 100049, China

*    Correspondence: baowenliao@caf.ac.cn; Tel.: +86-020-8702-8494

**Abstract:** Mangroves are distributed in the transition zone between sea and land, mostly in tropical and subtropical areas. They provide important ecosystem services and are therefore economically valuable. DNA barcoding is a useful tool for species identification and phylogenetic reconstruction. To evaluate the effectiveness of DNA barcoding in identifying mangrove species, we sampled 135 individuals representing 23 species, 22 genera, and 17 families from Zhanjiang, Shenzhen, Huizhou, and Shantou in the Guangdong province, China. We tested the universality of four DNA barcodes, namely *rbcL*, *matK*, *trnH-psbA*, and the internal transcribed spacer of nuclear ribosomal DNA (ITS), and examined their efficacy for species identification and the phylogenetic reconstruction of mangroves. The success rates for PCR amplification of *rbcL*, *matK*, *trnH-psbA*, and ITS were 100%, 80.29% ± 8.48%, 99.38% ± 1.25%, and 97.18% ± 3.25%, respectively, and the rates of DNA sequencing were 100%, 75.04% ± 6.26%, 94.57% ± 5.06%, and 83.35% ± 4.05%, respectively. These results suggest that both *rbcL* and *trnH–psbA* are universal in mangrove species from the Guangdong province. The highest success rate for species identification was 84.48% ± 12.09% with *trnH-psbA*, followed by *rbcL* (82.16% ± 9.68%), ITS (66.48% ± 5.97%), and *matK* (65.09% ± 6.00%), which increased to 91.25% ± 9.78% with the addition of *rbcL*. Additionally, the identification rate of mangroves was not significantly different between *rbcL* + *trnH-psbA* and other random fragment combinations. In conclusion, *rbcL and trnH-psbA* were the most suitable DNA barcode fragments for species identification in mangrove plants. When the phylogenetic relationships were constructed with random fragment combinations, the optimal evolutionary tree with high supporting values (86.33% ± 4.16%) was established using the combination of *matK* + *rbcL* + *trnH-psbA* + ITS in mangroves. In total, the 476 newly acquired sequences in this study lay the foundation for a DNA barcode database of mangroves.

**Keywords:** mangroves; DNA barcoding; species identification; phylogenetic relation

---

## 1. Introduction

In 2003, Hebert [1] proposed a novel DNA barcoding technology to expedite the process of species identification. Around 2005, the concept of DNA barcoding was introduced into botanical research [2,3]. In 2009, the CBOL(Consortium for the Barcode of Life) Plant Working Group initially identified and recommended the use of the chloroplast-derived DNA barcode fragments *rbcL* and *matK* [4]. In addition, *trnH-psbA*, the internal spacer region of the chloroplast gene, and the ITS region of the nuclear genome were also investigated [5–7]. Further research is required to compare DNA barcode

fragments to test their efficacy for species identification [8–11]. Previous studies on the DNA barcoding of plants have mainly focused on tropical and subtropical forests [12–15]. In addition, sequences obtained from DNA barcode fragments can also be used to reconstruct the phylogenetic relationships of specific biological groups, and this has become a new research hotspot in recent years [16–18]. This research promotes the integration of phylogenetic analysis, ecology, and barcoding technology and develops our understanding of evolutionary biology and other related disciplines [19–22].

Mangroves, in the broad sense, are woody plant communities growing in tropical and subtropical intertidal zones, which play an important role in maintaining the ecological balance of coastal zones [23,24]. There are 84 species of 24 genera and 16 families of mangrove plants in the world, including 70 species of 16 genera and 11 families of true mangrove plants and 14 species of eight genera and five families of semi-mangrove plants [25]. There are 25 species of mangrove plants from Guangdong, including 16 true mangroves and 9 semi-mangroves [26]. There are many similar species of mangroves, and their distribution areas are overlapped. It is difficult to identify similar mangrove species using the external morphology of plants, such as *Sonneratia alba* J. Smith, *Sonneratia caseolaris* (L.) Engl., *Sonneratia Hainanesis* Ko, E. Y. Chen et W. Y. Chen, *Sonneratia paracaseolaris* W. C. Ko et al. or *Bruguiera sexangula* (Lour.) Poir and *Bruguiera sexangula* (Lour.) Poir. var. rhynchopetala Ko. Moreover, it is hard to understand the evolutionary relationship between mangrove species with traditional classification methods. DNA barcoding in animals [27,28], insects [29,30], tropical and subtropical plants [19,31], and microorganisms [32,33] have achieved reliable reconstructions of evolutionary relationships, successfully identified species of the same genus, and discovered new species or cryptic species. In this study, we aimed to investigate the universality of DNA barcoding in the mangrove flora, which is in the transition zone between land and water, and to construct the phylogenetic trees of mangrove flora, to provide a scientific basis for the conservation of mangrove biodiversity.

## 2. Materials and Methods

### 2.1. Plant Material

In this study, the main distribution areas of mangroves in Guangdong were selected, namely the mangrove protection areas in Shenzhen, Huizhou, and Shantou in Eastern Guangdong and Zhanjiang in Western Guangdong. Specific sampling location information is shown in Table 1. According to the DNA barcode sample collection specifications of Gao [34], two to three individuals of each mangrove species were sampled. This involved taking fresh leaves and buds to facilitate the extraction of DNA molecular materials. Each individual sampled was more than 20 m apart. The test material was dried with silica gel after collection. A total of 135 individuals of mangrove plants were collected. Based on expert (LIAO and LI) identification, the individuals were classified into 23 species (including two mangrove companion species) of 22 genera belonging to 17 families. The number of true mangrove and semi-mangrove species sampled in this study accounted for 84% of the 25 species of mangrove plants in Guangdong province [26]. Among them, *B. sexangula* (Lour.) Poir and *B. sexangula* (Lour.) Poir. var. rhynchopetala Ko, which were introduced by humans, are almost extinct, so they were not sampled.

**Table 1.** The information of mangrove samples collected in the Guangdong province.

| Site | Collection Places | Longitude | Latitude | Numbers of Tree Species |
|---|---|---|---|---|
| Zhanjiang | Baitoupo | 109°45′38″ | 21°34′03″ | 16 |
| | Fengdi | 109°46′21″ | 21°32′36″ | |
| Shenzhen | Mangrove reserve | 114°00′33″ | 22°31′33″ | 12 |
| Huizhou | Yanzhou island | 114°56′17″ | 22°43′47″ | 14 |
| | Xiezhouwan | 114°47′00″ | 22°49′40″ | |
| | Haozhaolou | 114°52′02″ | 22°45′58″ | |
| Shantou | Su aiwan | 116°43′36″ | 23°19′16″ | 9 |
| | Waishahe | 116°48′11″ | 23°23′09″ | |
| | Yifengxi | 116°53′11″ | 23°32′45″ | |

### 2.2. DNA Extraction and Sequence Analysis

The DNA of mangrove plants was extracted from silica gel-dried leaf material following a modified version of the cetyltrimethyl ammonium bromide (CTAB) protocol of DoyleandDoyle [35]. According to the recommendation of the international union for biological barcoding [4] and previous studies [13,36,37] on regional plant DNA barcoding, a total of four molecular sequences including chloroplast *rbcL*, *matK*, *trnH–psbA*, and nuclear genome ITS were selected for use as amplification fragments. Referrals to the PCR system recommended by the plant working group of the international DNA barcode alliance for life, optimization, and adjustment were made. Primer information and amplification procedures are provided in Table 2. All amplification products were sent to Guangzhou after gel electrophoresis detection for complete sequencing; BLAST (Basic Local Alignment Search Tool) searches were performed using GenBank for the sequences obtained after bidirectional sequencing of the four fragments. If significant inconsistencies were found between the sequences and the original species, reasons were found and reextracted or reconfirmed by consulting experts, until the BLAST results of the sequences and the original species were of the same genus or family. SeqMan 5.00 (DNASTAR package, Madison, WI, USA) was used to splice and proofread the obtained sequences. The sequences were aligned in Geneious 11.1.3 (Biomatters Ltd, Auckland, New Zealand) using the MAFFT (Multiple Alignment using Fast Fourier Transform) algorithm with the default parameters.

**Table 2.** The primers used to amplify DNA barcodes and the amplification protocol.

| DNA Fragment | Primers | Sequences (5′–3′) | References | Amplification Protocol |
|---|---|---|---|---|
| *rbcL* | *rbcL*a_F | ATGTCACCACAAACAGAGACTAAAGC | Kress [19] | 94 °C 3 min; 94 °C 30 s, 55 °C 1 min, 72 °C 1 min, 35 cycles; 72 °C 7 min |
| | *rbcL*a_R | GTAAAATCAAGTCCACCRCG | | |
| *matK* | Kim_3F | CGTACAGTACTTTTGTGTTTACGAG | Kim, unpublished | 94 °C 3 min; 94 °C 45 s, 51 °C 45 s, 72 °C 1 min, 35 cycles; 72 °C 7 min |
| | Kim_1R | ACCCAGTCCATCTGGAAATCTTGGTTC | | |
| *trnH-psbA* | psbA3 | GTTATGCATGAACGTAATGCTC | Sang [38] | 94 °C 3 min; 94 °C 30 s, 55 °C 1 min, 72 °C 1 min,35 cycles; 72 °C 7 min |
| | trnH05 | CGCGCATGGTGGATTCACAATCC | Tate [39] | |
| ITS | ITS-leu1 | GTCCACTGAACCTTATCATTTAG | Urbatsh | 94 °C 3 min; 94 °C 30 s, 55 °C 1 min, 72 °C 1 min, 35 cycles; 72 °C 7 min |
| | ITS4 | TCCTCCGCTTATTGATATGC | White [40] | |

ITS: The internal transcribed spacer of nuclear ribosomal DNA.

### 2.3. Statistical Analysis

PCR amplification success rates and sequencing success rates were calculated following Kress [19]. The success rate of PCR amplification refers to the percentage of successful individuals of a segment in all individuals of the segment. BLAST was used to evaluate the efficacy of the species identification method. Firstly, a local database was established for the four DNA fragments in Geneious 11.1.3 [41], and all sequence comparisons were saved as *. fasta files to adjust the sequence direction and clear the gap between sequences. BLAST-2.7.1+ (https://www.ncbi.nlm.nih.gov/ package, National Center for Biotechnology Information, USA) was used to compare each sequence with all sequences in the database, and the percentage of identical sites was used as the quantification standard. If the minimum value of the identical sites of the same species was greater than the value between individuals of all other species, then we considered that the sequence of this species had been accurately identified. The success rate of identifications was determined by multiplying the percentage of species successfully identified by the success rate of sequencing the segment. Joint fragment identification is the result of accumulation on a single fragment [13,41].

## 3. Results

### 3.1. Sequence Universality

Sequence statistics were calculated for 135 individuals of 23 species of mangrove plants. The results (Table 3) showed a total of 496 DNA barcode fragments were obtained, with a sequence acquisition rate of 88.15% (476 divided by 540). Among them, a total of 118 sequences of mangrove plants from the Zhanjiang mangrove reserve were obtained. The highest success rate for PCR amplification were with *rbcL* and *trnH-psbA* fragments, followed by ITS and *matK*. The highest sequencing success rate was 100% with the *rbcL* fragment, followed by *trnH-psbA*, ITS, and *matK*.

**Table 3.** The success rates of PCR amplification and sequencing of the four barcoding fragments in the four mangrove forests.

| Plot | DNA Fragment | PCR Amplification Success Rate (%) | Sequencing Success Rate (%) | Individuals Resolution Rate |
|---|---|---|---|---|
| Zhanjiang | *rbcL* | 100 | 100 | 35 |
| | *matK* | 71.43 | 65.71 | 23 |
| | *trnH-psbA* | 100 | 88.57 | 31 |
| | ITS | 94.29 | 82.86 | 29 |
| Shenzhen | *rbcL* | 100 | 100 | 36 |
| | *matK* | 80.56 | 77.78 | 28 |
| | *trnH-psbA* | 100 | 97.22 | 35 |
| | ITS | 94.44 | 88.89 | 32 |
| Huizhou | *rbcL* | 100 | 100 | 40 |
| | *matK* | 77.5 | 77.5 | 31 |
| | *trnH-psbA* | 97.5 | 92.5 | 37 |
| | ITS | 100 | 82.5 | 33 |
| Shantou | *rbcL* | 100 | 100 | 24 |
| | *matK* | 91.67 | 79.17 | 19 |
| | *trnH-psbA* | 100 | 100 | 24 |
| | ITS | 100 | 79.17 | 19 |
| Total | | | | 476 |

A total of 131 sequences of mangrove plants from the mangrove reserve of Shenzhen were obtained. The highest amplification success rates were with both *rbcL* and *trnH-psbA*, followed by ITS and *matK*. The highest sequencing success rate was 100% with *rbcL*, followed by *trnH-psbA*, ITS, and *matK*. A total of 141 sequences of mangrove plants from the Huizhou mangrove reserve were obtained. The highest amplification success rates were with *rbcL*, followed by ITS, *trnH-psbA*, and *matK*. The highest sequencing success rate was 100% with *rbcL*, followed by *trnH-psbA*, ITS, and *matK*. A total of 86 sequences of mangrove plants from the Shantou mangrove reserve were obtained. The amplification success rates were 100% with *rbcL*, *trnH-psbA* or ITS, and 91.67% with *matK*. The highest sequencing success rate was 100% for *rbcL*, followed by *trnH-psbA*, ITS, and *matK*.

### 3.2. Species Delimitation

The true mangroves and semi-mangroves distributed in Guangdong are all single-genus and single-species, except *Sonneratia* and *Bruguiera*. Therefore, only the success rates of species-level identification are discussed here. BLAST results showed that the highest success rate of species identification was 84.48% ± 12.09% with *trnH-psbA*, followed by 82.16% ± 9.68% with *rbcL*, 66.48% ± 5.97% with ITS, and 65.09% ± 6.00% with *matK* (Figure 1). However, more species were successfully identified with *rbcL* fragments than with *trnH-psbA* fragments in any region, except for Shantou. The main difference was that *trnH-psbA* could accurately distinguish *S. caseolaris* (L.) Engl. and *Sonneratia apetala* Buch.-Ham, but *rbcL* could not.

According to the statistical analysis of different combinations of multiple fragments, the identification success rate of *rbcL + trnH-psbA* was 91.25% ± 9.78%, which increased to 91.88% ± 8.62% with the addition of ITS. The success rate of species identification with all fragments was consistent with that of *rbcL + trnH-psbA + matK* (94.38% ± 4.48%). These results suggest that the combination of certain fragments can increase the success rate of barcoding for species identification.

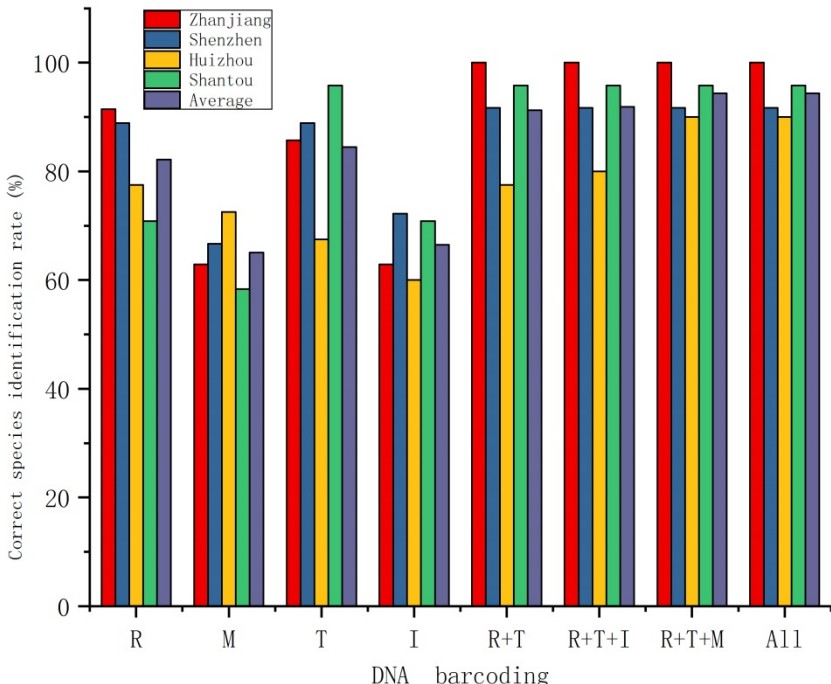

**Figure 1.** Mangrove species discrimination rate of all single and multi-DNA barcoding fragments. R, *rbcL*; M, *matK*; T, *trnH–psbA*; I, ITS; All, the database consisted of all species.

*3.3. Phylogenetic Trees*

Phylogenetic trees were constructed in MEGA6.0 (Tamura, Stecher, Peterson, Filipski, and Kumar) using the neighbor-joining (NJ) method based on the Kimura'2-parameter model. Phylogenetic trees were constructed with the individual or random combined fragments and the average node support rate was calculated. The results showed that the highest average node support rate of phylogenetic trees using the combination *rbcL + matK + trnH-psbA* + ITS was 86.33% ± 4.16% (as shown in Figures 2–5). Phylogenetic trees were fan-shaped, with one branch of the same or similar species. The average node support rate for mangrove phylogenetic trees in the four regions was 89.66% ± 18.50% in Zhanjiang, 88.49% ± 17.25% in Huizhou, 86.85% ± 15.60% in Shenzhen, and 80.33% ± 19.89% in Shantou.

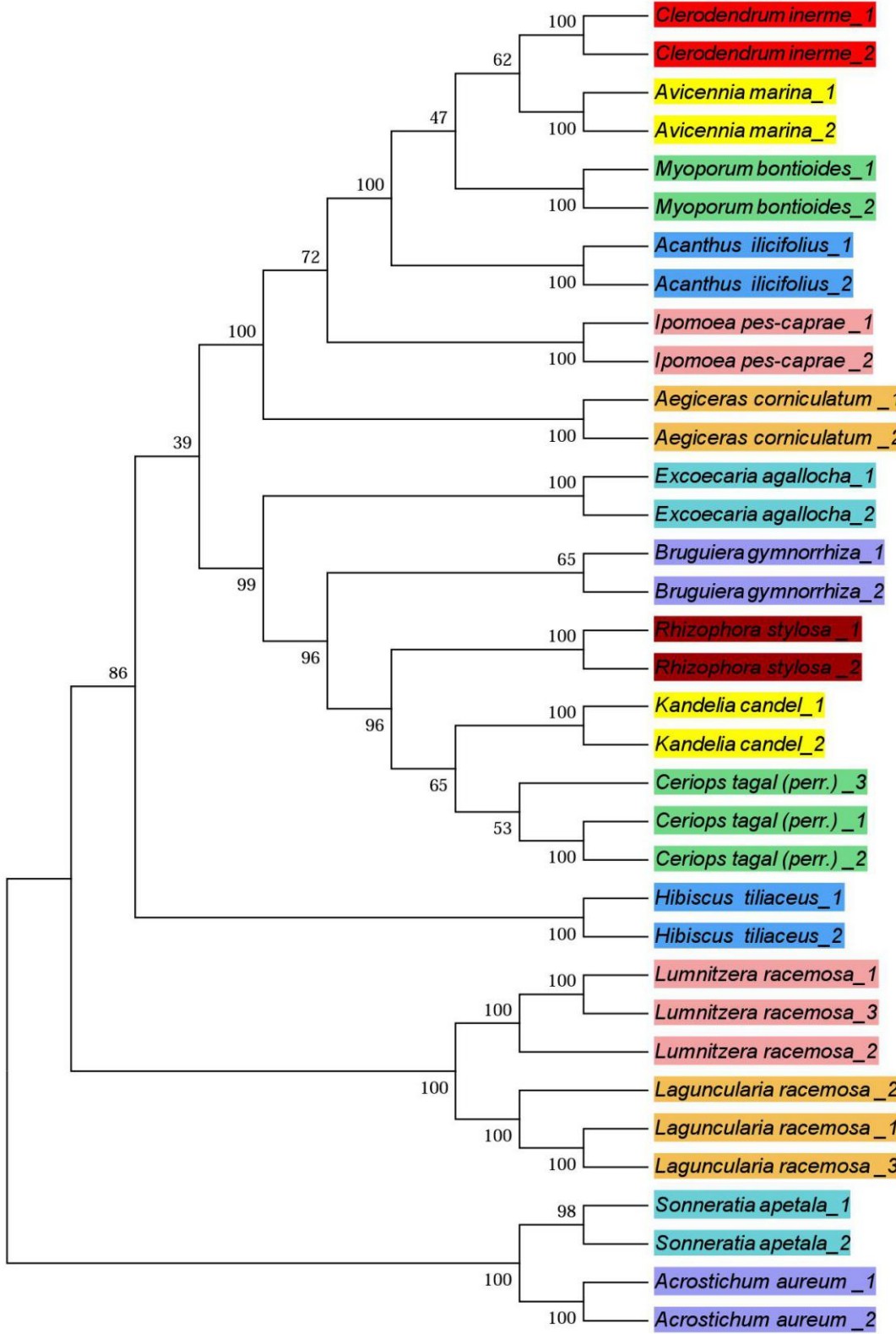

**Figure 2.** The phylogenetic tree of mangroves in Zhanjiang using fragment of *matK* + *rbcL* + *trnH-psbA* + ITS.

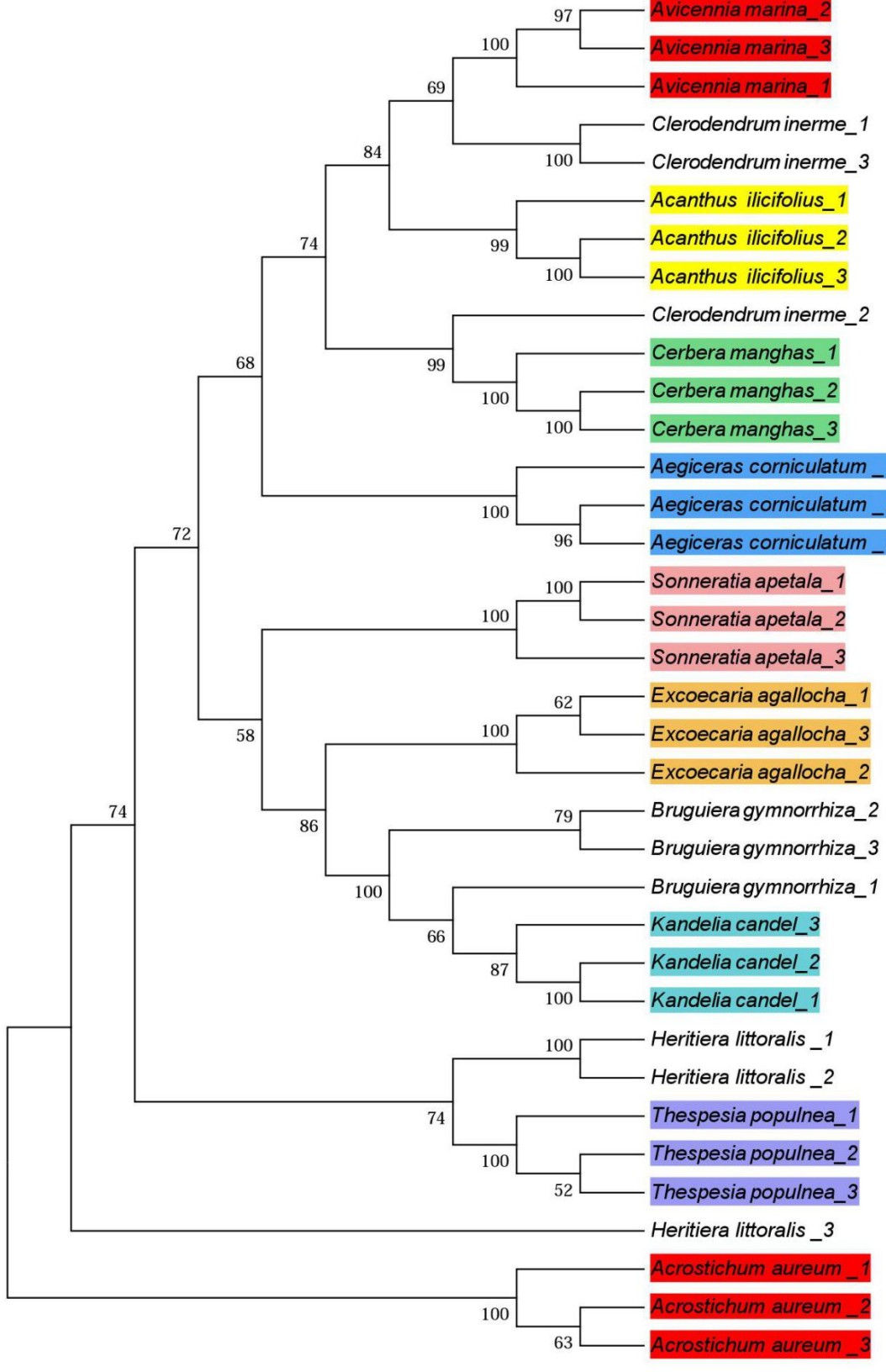

**Figure 3.** The phylogenetic tree of mangroves in Shenzhen using fragment of *matK + rbcL + trnH-psbA* + ITS.

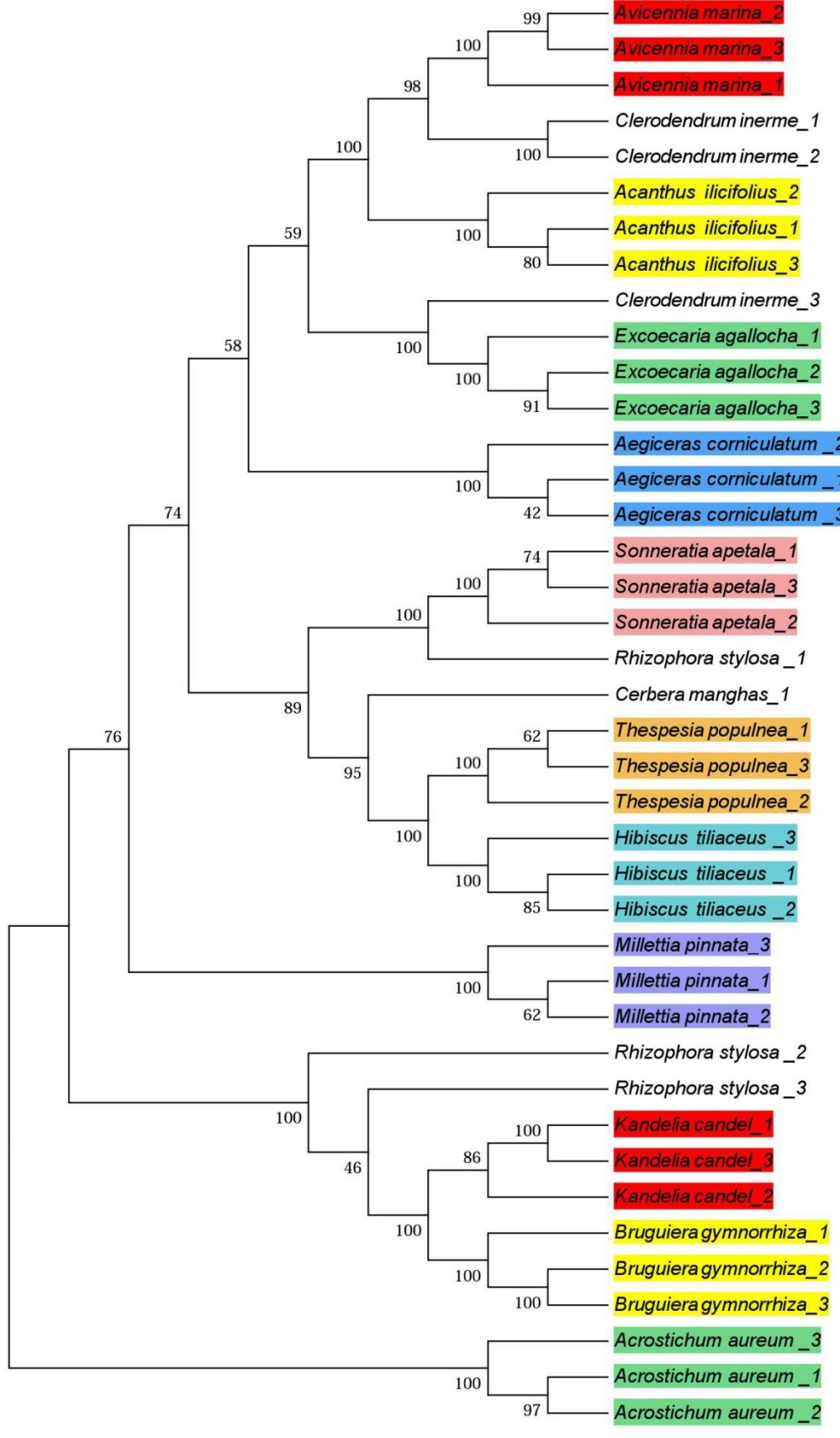

**Figure 4.** The phylogenetic tree of mangroves in Huizhou using fragment of *matK* + *rbcL* + *trnH-psbA* + ITS.

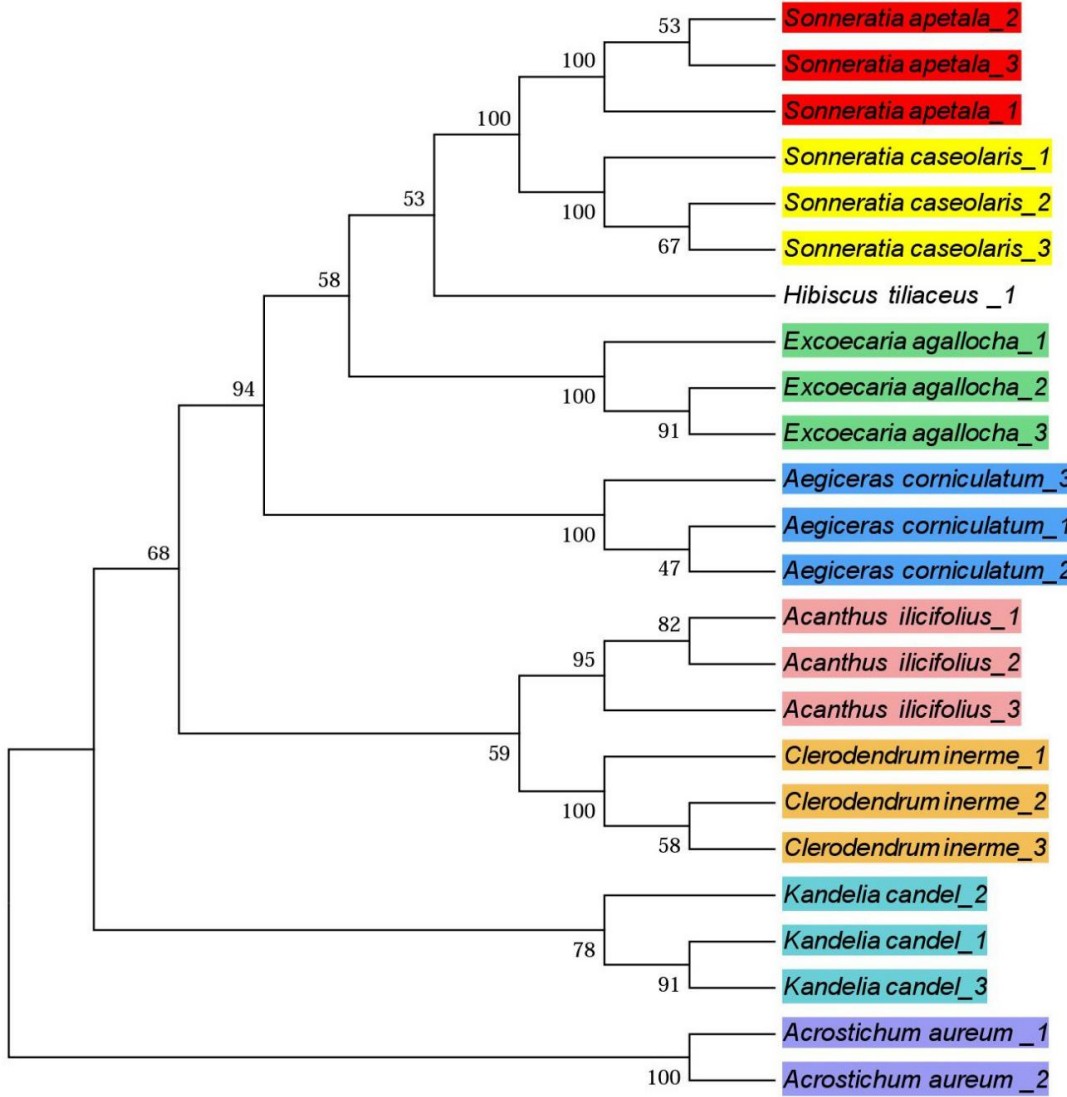

**Figure 5.** The phylogenetic tree of mangroves in Shantou using fragment of *matK* + *rbcL* + *trnH-psbA* + ITS.

## 4. Discussion

Our study investigated mangrove plants in the Guangdong province using DNA barcoding technology. The purpose of this study was to evaluate the performance of DNA barcoding in terms of primer universality, successful identification rate, and phylogenetic tree construction.

### 4.1. The Universality of DNA Barcoding in Mangrove Communities

The success rate of PCR amplification and the sequencing of *rbcL* fragments in core barcodes of mangrove DNA samples reached 100%. Compared with Kress [19], our results showed higher universality and success rates, and were the same as those of Pei [42], with 90%–100% in the forest plant communities in tropical and subtropical regions. This indicates that *rbcL* has the fewest numbers of variable sites and that the selected primer sequence has strong universality. Therefore, *rbcL* is recommended as an effective fragment for DNA barcoding in mangroves. The success rate of another core fragment, *matK*, was the lowest of the four fragments. With this fragment, the amplification or sequencing of *Ceriops tagal* (perr.) C. B. Rob. and *Kandelia candel* (L.) Druce were not successful, and only a small number of individuals were successfully sequenced in, e.g., *Bruguiera gymnorrhiza* (L.) Poir., *Acrostichum aureum* L., and *Rhizophora stylosa* Griff., indicating a lack of universality for this

fragment compared to *rbcL*. This may be due to the existence of single nucleotide repeat sequences. The amplification rate of *matK* fragments in previous studies was lower, e.g., 68.18% [43], as was the success rate, e.g., 64% [14]. Similarly low results were reported in other studies of Liu [13], Lu [44], and Wei [45]. Generally, in this situation, the number of primers increases, and the procedure is repeated multiple times.

The barcode *trnH-psbA* has high universality, and the amplification rate and sequencing success rate can be 99.38% ± 1.25% and 94.57% ± 5.06%, respectively, with only one pair of primers, which is second only to the *rbcL* fragment. This fragment can also successfully distinguish *Sonneratia*. The success rate of the nuclear gene ITS amplification was 97.18% ± 3.25%; the amplifications of *A. aureum*, *B. gymnorrhiza*, *C. tagal*, and *R. stylosa* were not successful. The sequencing success rate of the nuclear gene ITS was 83.35% ± 4.05%, and only a few individuals were successfully sequenced in *K. candel* and *Acanthus ilicifolius* L. Previous studies, such as that of Tripathi [46], showed that the sequencing success rate of ITS sequences in tropical forest species in India was 62.0%. Kang [47] showed that the success rate of ITS sequencing in Hainan tropical cloud forests was 47.20% ± 5.76%. In conclusion, the *rbcL* fragment and *trnH-psbA* fragment are recommended for the amplification and sequencing of mangrove plants.

### 4.2. Species Identification Ability

CBOL Plant Working Group recommend *rbcL* and *matK* for the core DNA barcoding for plants. Burgess [41] found that the core barcode can successfully identify 93% of species in the temperate flora of Canada, and de Vere [48] found that *rbcL* + *matK* can identify 69.4%–74.1% of flowering plants in Wales, UK. Kress [19] studied 296 woody species in Panama and found that the species identification rate of *matK* + *rbcL* was as high as 98%. In this study, the species identification rate of the core barcode *rbcL* + *matK* combination fragment can reach 90.56% ± 6.94%. This indicates that the core barcode is effective for species identification and is consistent with previous research results. However, the identification rate of mangroves decreased when only one of them was used. Either *rbcL* or *trnH-psbA* had a higher identification rate of mangroves than that of *matK*. Additionally, the rate for *rbcL* + *trnH-psbA* was higher than that for *rbcL* + *matK*. Furthermore, the identification rate of mangroves had no significant differences between *rbcL* + *trnH-psbA* and other random fragment combinations.

Although the number of bases of *trnH-psbA* varies greatly among different plant groups, a large number of insertions and deletions make sequencing difficult, as well as the existence of single-nucleotide repeats and other special structures in some groups. In this study, the success rate of *trnH-psbA* sequencing was second only to *rbcL*. At the same time, this fragment was able to successfully identify *S. caseolares* and *S. apetala*. The species identification rate of *trnH-psbA* was the highest among the four single fragments, Gonzalez [12] and Tripathi [46] also showed that *trnH-psbA* is one of the most promising barcodes for species identification. This suggests that *trnH-psbA* may act as a complementary fragment to *rbcL*.

Although studies by Kress [19], Sass [49], and Li [7] support the incorporation of ITS into the core barcode of plant DNA, the supplementary ITS fragment was found to be of low universality in the present study, and the species identification rate, when used singly, was 66.48% ± 5.97%. However, the ITS fragment could identify *Laguncularia racemosa* C.F.Gaertn but not *rbcL* and *trnH-psbA*. In addition, the ITS region of the nuclear genome can provide more genetic information from parents than the chloroplast genes. Given the above, *rbcL* and *trnH-psbA* were effective fragments for mangrove identification, and ITS fragment could be used for specific mangroves.

### 4.3. Phylogenetic Trees

*rbcL* + *matK* + *trnH-psbA* fragments had been used to construct phylogenetic trees in different localities, e.g., Barro Colorado Island [19], the Dinghu mountain forest [50], the Ailao mountain forest [44], and tropical cloud forests in Hainan [47]. However, the average node support for phylogenetic trees of mangrove species in these four subtropical regions constructed with *rbcL* +

*matK* + *trnH-psbA* + ITS was 5.59% higher than that with *rbcL* + *matK* + *trnH-psbA*, which saw no significant difference. In addition, in Figures 2–5, most of the same species were clustered into one branch, also indicating that DNA barcoding can be used to identify species. Due to the different geographical locations of the mangroves, environmental conditions are also different (Table 1), which leads to differences in selective pressure. This could lead to differences in the evolutionary trajectory of mangrove species in the four regions. For example, community composition was observed to be different at different locations, whereby true mangrove species gradually decreased from south to north.

## 5. Conclusions

The results of the present study suggest that *rbcL* and *trnH-psbA* have high success rates for amplification and sequencing, indicating that these two barcodes are common in mangrove species. In terms of species identification, these fragments were relatively successful compared to the other fragments tested. The phylogenetic trees of mangrove plants constructed with a combination of *rbcL* + *matK* + *trnH-psbA* + ITS had the highest node support rate. Due to the low efficiency of *matK* fragment amplification and identification in mangrove plants. And the identification success rate of *rbcL* was higher than that of *trnH-psbA*, except for Shantou region, where the fragment of *trnH-psbA* can be used to identify specific species which cannot be identified by *rbcL*. Thus, this study concluded that *rbcL* and *trnH-psbA* were the most suitable DNA barcode fragments for species identification in mangrove plants.

Data collection for mangrove DNA barcoding in the Guangdong province is ongoing. A total of 476 sequences were obtained, from 135 individuals of 23 species of mangrove plants, accounting for 55.26% of the mangrove plants in China (21/38). In future research, the sampling range can be further expanded to include the DNA barcoding of other mangrove tree species and mangrove companion plants, to build a complete and high-coverage mangrove plant DNA barcoding database.

**Author Contributions:** Conceptualization, B.L. and M.L.; methodology, F.W.; software, F.W., Y.X.; formal analysis, F.W., Y.X.; investigation, F.W., B.L., M.L., X.S.; supervision, B.L., M.L.; writing, F.W.

**Funding:** This research was funded by the Ministry of Science and Technology of China (No. 2017FY100705 and No. 2017FY100700), and the National Natural Science Foundation of China (No. 31570594).

**Acknowledgments:** The authors would like to thank Jiang, Z.M. and Xu, Y.W. for collecting material, and additional help. Our sincere thanks are extended to Yan, H.F. for providing a scientific research platform.

**Conflicts of Interest:** The authors declare no conflict of interest.

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
