# Peer review of "DNA Barcoding Analysis and Phylogenetic Relation of Mangroves in Guangdong Province, China"

_forests, doi:10.3390/f10010056_

Round 1
Reviewer 1 Report
The authors report the use of four sequences (3 chloroplast and one nuclear ITS fragments) to serve as DNA barcoding for the identification of mangrove species. Their study focused on 23 species in 15 families from Guangdong province in China. In addition, they constructed phylogenetic trees based on the different sequences.
This should be a brief communication and as it is presented is longer than necessary:
1) I do not see the value in the phylogenetic analyses. As mentioned in the text, the species are drawn from 15 families. Since the representation of each family is so minimal, no evolutionary conclusions can be drawn from the trees, contrary to the statement on lines 258-260 page 12. This would eliminate the repetition of different trees using combinations of the sequences.
2) There is considerable repetition in the text of data presented in tables. The text could be simplified by describing only the most important points and referring to the tables.
3) The combination of four sequences appears to have performed well, although PCR and sequencing tended to be low for matK. This is mentioned in the text, but I wonder why Fig 1 does not show the results of combinations of the three other sequences without matK.
4) It would be useful to include some justification of the necessity of DNA barcoding of mangrove species.
A minor point;
In the Introduction, the authors state that molecular studies of mangroves are rarely reported (lines 58-60). This is not true, since there have been many molecular, genetic and evolutionary studies of mangroves.
Author Response
Point 1: I do not see the value in the phylogenetic analyses. As mentioned in the text, the species are drawn from 15 families. Since the representation of each family is so minimal, no evolutionary conclusions can be drawn from the trees, contrary to the statement on lines 258-260 page 12. This would eliminate the repetition of different trees using combinations of the sequences..
Response 1: I add the phylogenetic analyses,eg:(1) Since mangrove communities have many similar species and endemic species, the phylogenetic tree constructed by the APG online system (constrained tree) is difficult to reflect the phylogenetic relationships in the mangrove communities. Phylogenetic trees constructed by DNA sequences (non-constrained tree) in this study are able to clearly cluster the closely related species and separate distantly related species in mangrove communities (Figs 2–5)
In addition,(1) Most mangrove plants are a single species in the plant genes. (2) When the number of species is small, there is no significant difference in the support rate of phylogenetic tree with the combination of four fragments or R+M+T. In addition, the small number of species is also the deficiency of this study. But we had tried our best to collect 84% of all species of mangrove in Guangdong province. In the future, the number of mangrove plant species can be increased to compare the difference between the two fragments combination in phylogenetic tree.
Point 2: There is considerable repetition in the text of data presented in tables. The text could be simplified by describing only the most important points and referring to the tables.
Response 2: The text was simplified by describing only the most important points and referring to the tables. For example, line 121-124:
A total of 118 sequences of mangrove plants from the Zhanjiang mangrove reserve were obtained. The highest success rate for PCR amplification were with rbcL and trnH-psbA fragments, followed by ITS and matK. The highest sequencing success rate was 100% with the rbcL fragment, followed by trnH-psbA, ITS and matK.
Point 3:The combination of four sequences appears to have performed well, although PCR and sequencing tended to be low for matK. This is mentioned in the text, but I wonder why Fig 1 does not show the results of combinations of the three other sequences without matK.
Response 3: The identification rate of combinations of the three other sequences without matK is 91.88%±8.62, and I modify figure 1.
Point 4: It would be useful to include some justification of the necessity of DNA barcoding of mangrove species.
Response 4:The necessity of DNA barcoding of mangrove species was added in introduction. Eg: There are many similar species of mangroves, and the distribution of them are overlapped.
It is difficult to identify similar mangrove species by using external morphology of plants, such as Sonneratia slba, S. caseolaris, S. hainanesis, S. paracaseolaris or Bruguiera sexangula and B. Sexangula. And it is hard to understand the evolutionary relationship between mangrove species with traditional classification methods. In order to resolve these problems, we start this research.
A minor point; In the Introduction, the authors state that molecular studies of mangroves are rarely reported (lines 58-60). This is not true, since there have been many molecular, genetic and evolutionary studies of mangroves.
Thanks , After considering,this sentence is not closely related to the text, so i deleted it。

Reviewer 2 Report
You used a group of plants important for special ecosystems to identify the species and look for the phylogenetic relationships between this mixture of families, genera and species. Overall this therefore is an interesting paper to a broader community with as well interests in mangroves as in barcoding systems in general.
Nevertheless, I have some questions and remarks that I would like to be answered or changed:
Which families, genera, species were used? Only a number how many is given but not which. You cannot assume that all readers of your article know which families, genera and species are naturally distributed in Guandong area. But, you have to assume that your readers are interested in knowing which species you investigated
Table 3: trnH-psbA is not given in full but only as “t” without explanation. Either give the full name or explain it in the title of the table or a legend.
Page 5, line 144: delete “As” at the beginning of the sentence
Page 6, line 152-154: Why a combination of matK + rbcL, whereas beforehand it is said that rbcL and trnH-psbA have the highest species resolution? Which statistical analysis gave you the result that nevertheless matK + rbcL is the most promising combination?
Page 6, line 161 and following: A few lines earlier you describe that the addition of ITS decrease the success rate for species identification slightly. What are the reasons that you use all four fragments for construction of the phylogenetic trees?
You investigated 23 species, but your phylogenetic trees show a maximum of 14 species – where are the others and again, which are the others?
Minor comment:
Please decide whether you write the cp fragments in Italic or not but do not mix it up.
Author Response
Point 1: Which families, genera, species were used? Only a number how many is given but not which. You cannot assume that all readers of your article know which families, genera and species are naturally distributed in Guandong area. But, you have to assume that your readers are interested in knowing which species you investigated.
Response 1: Wo can find the which families, genera and species are naturally distributed in Guandong area at reference 26. In addition, The specific species of each region are listed in figure 2-5, Considering the length of the article, I take the sample collection information as supplementary materials
NO | Family | Species |
1 | Acrostichaceae | Acrostichum aureum |
2 | Euphorbiaceae | Excoecaria agallocha |
3 | Sonneratiaceae | Sonneratia caseolares |
4 | Sonneratia apetala | |
5 | Rhizophoraceae | Bruguiera gymnoihiza |
6 | Ceriops tagal | |
7 | Kandelia candel | |
8 | Rhizophora stylosa | |
9 | Combretaceae | Lumnitzera racemosa |
10 | Laguncularia racemosa | |
11 | Myrsinaceae | Aegiceras corniculatum |
12 | Verbenaceae | Avicennia marina |
13 | Clerodendrum inerme | |
14 | Acanthaceae | Acanthus ilicifolius |
15 | Leguminosae | Pongamia pinnata |
16 | Malvaceae | Hibiscus tilisceus |
17 | Thespesia populnea | |
18 | Sterculiaceae | Heritiera littoralis |
19 | Apocynaceae | Cerbera manghas |
20 | Bignoniaceae | Dolichandron spathacea |
21 | Compositae | Pluchea indica |
22 | Myoporaceae | Myoporum bontioides |
23 | Convolvulaceae | Ipomoea pes-caprae |
Point 2: Table 3: trnH-psbA is not given in full but only as “t” without explanation. Either give the full name or explain it in the title of the table or a legend.
Response 2: I change “t ”to “trnH-psbA”
Point 3:Page 5, line 144: delete “As” at the beginning of the sentence
Response 3: I delete it
Point 4: Page 6, line 152-154: Why a combination of matK + rbcL, whereas beforehand it is said that rbcL and trnH-psbA have the highest species resolution? Which statistical analysis gave you the result that nevertheless matK + rbcL is the most promising combination?
Response 4: I modify it as following: According to the statistical analysis of different combinations of multiple fragments, the identification success rate of rbcL + trnH-psbA was 91.25% ± 9.78%, which increased to 91.88% ± 8.62% with the addition of ITS. The success rate of species identification with all fragments was consistent with that of rbcL + trnH-psbA + matK (94.38% ± 4.48%).
Point 5:Page 6, line 161 and following: A few lines earlier you describe that the addition of ITS decrease the success rate for species identification slightly. What are the reasons that you use all four fragments for construction of the phylogenetic trees?
Response 5: The species identification rates of R+M+T and R+M+T+I all were the highest, 94.38% + 4.48%. In addition, the node support rate of the phylogenetic tree with 4 fragments was higher than that with 3 fragments, so 4 fragments were selected to construct the phylogenetic tree
Point 6:You investigated 23 species, but your phylogenetic trees show a maximum of 14 species – where are the others and again, which are the others?
Response 6: In this study, 23 species were sampled in 4 sampling sites, including 16 species in Zhanjiang, 12 species in Shenzhen, 14 species in Huizhou and 9 species in Shantou. Phylogenetic trees are built at every different regions.
Point 7: Please decide whether you write the cp fragments in Italic or not but do not mix it up.
Response7: I write the cp fragments in Italic

Round 2
Reviewer 1 Report
The authors have addressed all, but the first of my comments satisfactorily.
My first comment was the utility of showing phylogenetic trees. Contrary to the statement in the Discussion that the trees show evolutionary relationships, such a mixed group of species cannot address evolutionary relationships. There are too many gaps in terms of species and higher taxonomic orders for this to have any meaning. The authors state that the APG constrained tree is not very useful for the mangrove. However, mangroves form an ecological community including taxa from a wide spread of families. I think if the authors insist on including the trees they should remove any reference to evolutionary relationships and simply use the trees to show that divergent taxa can be identified using their barcoding primers
Author Response
Thank you for your comment. The purpose of my phylogenetic trees is to illustrate the identification results of DNA barcoding on species. It's really not appropriate to use them to illustrate the evolutionary relationship of species. I revised the paper as follows:
rbcL + matK + trnH-psbA fragments have been used to construct phylogenetic trees in different localities, e.g., Barro Colorado Island [19], Dinghu mountain forest [50], Ailao mountain forest [44], and tropical cloud forests in Hainan [47]. However, the average node support for phylogenetic trees of mangrove species in these four subtropical regions were constructed with rbcL + matK + trnH-psbA + ITS was 5.59% higher than that with rbcL + matK + trnH-psbA, which was no significant difference. In addition, in Figures 2-5, most of the same species are clustered into one branch, also indicating that DNA barcoding can be used to identify species.
Furthermore, the references in this paragraph are used for illustrating that the combinations of DNA barcode fragment had ever be used to construct phylogenetic trees, so i modified the expression in this paper. And I retain relevant references.